# First Molecular Identification of *Bulinus africanus* in Lake Malawi Implicated in Transmitting *Schistosoma* Parasites

**DOI:** 10.3390/tropicalmed7080195

**Published:** 2022-08-19

**Authors:** Mohammad H. Alharbi, Cynthia Iravoga, Sekeleghe A. Kayuni, Lucas Cunningham, E. James LaCourse, Peter Makaula, J. Russell Stothard

**Affiliations:** 1Department of Tropical Disease Biology, Liverpool School of Tropical Medicine, Liverpool L3 5QA, UK; 2Ministry of Health, Buraydah 52367, Saudi Arabia; 3MASM Medi Clinics Limited, Medical Society of Malawi (MASM), Lilongwe P.O. Box 1254, Malawi; 4Research for Health, Environment and Development (RHED), Mangochi P.O. Box 345, Malawi

**Keywords:** *Schistosoma haematobium*, urogenital schistosomiasis

## Abstract

The freshwater snail genus *Bulinus* plays a vital role in transmitting parasites of the *Schistosoma haematobium* group. A hybrid schistosome between *S. haematobium* and *S. mattheei* has been recently detected using DNA-based identification methods in school children along the Lake Malawi shoreline in Mangochi District. This finding raised the need for contemporary revaluation of local interactions between schistosomes and snails, with a particular focus on snail species within the *Bulinus africanus* group. In 2017 and 2018, malacological surveys sampled several freshwater sites in Mangochi District. Collected snails (*n* = 250) were characterised using cytochrome oxidase subunit 1 gene (*cox*1), with DNA barcoding of the ‘Folmer’ region and a rapid PCR-RFLP typing assay with double digestion with *Hae*III and *Sac*I restriction enzymes. DNA *cox*1 sequence analysis, with phylogenetic tree construction, suggested the presence of at least three *B. africanus* group taxa in Lake Malawi, *B. globosus,* alongside first reports of *B. africanus* and *B. angolensis,* which can be differentiated by PCR-RFLP methods. In addition, a total of 30 of the 106 *B. africanus* group snails (28.30%) were positive to the *Schistosoma*-specific screen using real-time PCR methods. This study provides new insight into the recent changes in the epidemiology of urogenital schistosomiasis as likely driven by a new diversity of *B. africanus* group snails within the Lake.

## 1. Introduction

The freshwater snail genus *Bulinus* plays a crucial role in the transmission of species in the *S. haematobium* group that causes schistosomiasis. Meanwhile, the group includes three human pathogens (*S. haematobium, S. guineensis* and *S. intercalatum*) and five others that could infect wild and domestic ruminants (*S. bovis, S. leiperi, S. curassoni, S. margrebowiei* and *S. mattheei*). The interaction and relationship between schistosomes and snails are very precise, and compatibility may vary over relatively small geographical ranges [1]. Recently, there was a detection of a hybrid schistosome between *S. haematobium* and *S. mattheei* in Mangochi District along Lake Malawi among local people [2]. There is a lack of certainty about the compatibility of this hybrid schistosome with local snail species in Malawi. Some *B. africanus* group species are playing a role in the life cycle of *S. haematobium*, *S. bovis* and *S. mattheei* which would each continue to drive the hybridisation of human and zoonotic schistosomes.

It is essential to have up-to-date data about the intermediate hosts for schistosomiasis control; these data include accurate snail identity, their roles in transmission infection and population map distributions [3]. Several molecular characterisation methods are currently available to be used in the molecular identification of freshwater snails, such as DNA sequence analysis. The mitochondrial region of cytochrome oxidase subunit 1 (*cox*1) sequences is commonly used for snail identification, making it a suitable comparison gene for general species identification and phylogenetic inferences [4]. Fortunately, the relatively high genetic variation of *Bulinus* snails, particularly in the *cox*1 gene, provides a sufficient marker to distinguish between populations and species whereas it may be very challenging to identify some populations or individual snails utilizing traditional taxonomy [5]. Restriction fragment length polymorphism (RFLP) is a simple DNA characterisation method that relies on fragment sizes as cut by the restriction enzymes [6]. PCR-RFLP assays have been used for *Bulinus* snail discrimination, particularly *B. africanus* group [7,8].

Understanding the epidemiology of schistosomes requires a reliable taxonomy of the genus *Bulinus* [7]. For example, some species from the *B. africanus* group are well known or suspected as the intermediate host for *S. haematobium, S. bovis* and *S. mattheei.* However, classification within *B. africanus* species group complex is probably the most problematic of the four *Bulinus* snail species groups, as genetically this group is very diverse and also very geographically widely spread [7]. In Malawi, for example, it was previously believed that the open shores of Lake Malawi were free from *S. haematobium* as *B. globosus* was absent, being only found in more marshy marginal shorelines [9]. Surprisingly, in 2001, Madsen et al. (2001) reported that *B. nyassanus*, a deep-water snail, from the *B. truncatus/tropicus* complex could produce cercariae [10]. This new epidemiological change linked this snail with ongoing urogenital schistosomiasis transmission in some parts of the lake shoreline.

Species identification within the *B. africanus* group can be unreliable when based exclusively on morphology. The shell is often environmentally plastic in shape and size, being subject to ecophenotypic variation [11]. Brown (1994) suggests that more populations and advanced application of molecular analysis are needed to distinguish between *Bulinus africanus* group species with greater confidence to help in understanding their distribution and their roles in disease transmission [12]. Some studies have investigated certain species from this group using molecular identification methods applied across several populations [5,7,13]. The findings obtained from these studies were interesting, and some results were unexpected which reflect the need for further molecular identification methods for snail characterisation. For example, Allan et al. (2017) conducted a genetic investigation finding that *B. angolensis* should be reclassified to *B. africanus* group instead of originally documented in *B. truncatus/tropicus* complex [13].

The populations of *B. africanus* group in Malawi and precisely those snails in and along Lake Malawi have not been thoroughly investigated at a molecular level. Interestingly, during two malacological surveys in 2017 and 2018 in Mangochi District, Malawi, *B. africanus* group was found, often at human water contact sites within the lake as well as in fringing water bodies outside the lake. Therefore, this study aims to investigate these populations at a molecular level and, by exploiting different molecular applications, our study aims to find an optimum method to distinguish between different species in Malawi including *B. africanus* group and *B. truncatus/tropicus* complex. Although this study focuses on *B. africanus* group in Lake Malawi, we included some known *Bulinus* species collected from within and outside the lake for direct comparisons.

The development of rapid identification techniques and the data provided in this study would contribute to following the recent changes in *Schistosoma* intermediate hosts in Lake Malawi. Moreover, it helps in the reliable monitoring of the altering dynamic between the parasite and its snail host. Further observations are needed to evaluate the epidemiological changes reported recently in Malawi and the potential concerns of these hybridisation changes.

## 2. Materials and Methods

### 2.1. Sample Survey

Two malacological surveys were carried out in the Mangochi District along the southern lake shoreline and nearby water bodies in November 2017 (14 locations) and May 2018 (43 locations). *Bulinus* species samples were collected from the vast majority of visited sites, plus details of each site such as Global Positioning System (GPS). Then, each *B. africanus* shell was identified based on its morphological characteristics [12]. Additional *B. globosus* samples collected by J. R. Stothard from other two districts in Malawi were included in RFLP and sequencing analysis for comparison; the samples were collected from a river in Blantyre and a pond in Chikhwawa District located in the southern region of Malawi (Figure 1).

### 2.2. Morphology Study

A total of 135 *B. africanus* group snails were photographed with a background of a piece of paper covered by 1 mm^2^ squares. Then Motic Image Plus 3.0 (Kowloon, Hongkong) was used for five measurements for each shell (Figure 2). Then, the data were recorded onto an Excel sheet.

### 2.3. Molecular Characterisation Analysis

First, *B. africanus* group shells were photographed with a background of a piece of paper covered by 1 mm^2^ squares. Then, snail tissue from the foot and head (about 2 mm) was cut, and then we placed them in a 24-well plate, having distilled water for approximately 5 min to diffuse out ethanol in the tissue. Next, DNA was extracted using the DNeasy Blood and tissue kit (Qiagen™, Hilden, Germany), following the instructions. The quantity and quality of the DNA of samples were tested using spectrophotometry (Nanodrop™, Wilmington, NC, USA). Later, extracted DNA was diluted to 10 ng/µL in elution buffer (AE). Fifty samples were randomly selected for sequence analysis using *cox*1 gene, and partial mitochondrial *cox*1 was amplified using BulCox1 and HCO2198 [7,14] in PCR reaction for sequence analysis. Amplifications were performed using Illustra puReTaq Ready-To-Go PCR beads (GE Healthcare, Chicago, IL, USA) in 25 μL total volume consisting of 0.2 μM of forward and reverse primers, 21 μL of nuclease-free water and 2 μL of DNA. As per the manufacturer’s instructions, successfully amplified products were purified using MiniElute PCR purification kit (Qiagen™) before being sent for sequencing. This step was also conducted for non-*B. africanus* group samples to validate the results from the RFLP rapid assay.

### 2.4. Restriction Digestion

The *cox1* fragments amplified from the ‘Folmer’ region in the snails were digested using *SacI* and *HaeIII* restriction enzymes in CutSmart^®^ Buffer (Hitchin, UK). Based on sequenced samples used as a control in this study, two species from *B. africanus* group were identified as *B. africanus* and *B. globosus*. Thus, we developed two markers (restriction enzymes) in the PCR-RFLP essay to distinguish between the identified species through the sequences generated. *SacI* (GAGCT^C) was expected to cut only *B. globosus* samples, while *HaeIII* (GG^CC) only cut *B. africanus*. The reaction contained 3.5 µL of nuclease-free water, 5 µL of DNA, 1µL of each enzyme and 4.5 µL of a buffer. Then, the mixture was incubated to digest 3 h at 37.5 °C before running on 1% agarose gel stained with SYBR™ Safe DNA gel stain at 100 V for 40 min. Additional *B. globosus* samples from the different areas were included in PCR-RFLP and sequencing analysis for comparison. Several samples identified as *B. truncatus/tropicus* complex from different sites were included as a negative control for amplification and PCR-RFLP to test the specificity of the primers and restriction enzymes used in this study.

### 2.5. Real-Time PCR Detection of Schistosomes within Snails

Real-time PCR to detect schistosome within snail was conducted to detect *Schistosoma* spp. within *B. africanus* group samples using Ssp48F (5′-GGT CTA GAT GAC TTG ATY GAG ATG CT-3′), Ssp124R (5′-TCC CGA GCG YGT ATA ATG TCA TTA-3′) and the probe Ssp78T(FAM-5′-TGG GTT GTG CTC GAG TCG TGGC-3′) as described by [15].

### 2.6. Statistical and Analytical Methods

Molecular data were analysed using Geneious Prime 2019 (Auckland, New Zealand), National Centre for Biotechnology Information (NCBI) for BLAST and alignment using MUSCLE (Mill Valley, CA, USA) to investigate molecular identification and sequence variation.

### 2.7. Phylogenetics Analysis

Evolutionary analyses were conducted in MEGA X (Philadelphia, PA, USA) to create a maximum likelihood and neighbour-joining trees based on the sequence generated of a fragment from the *cox*1. Based on the best evolutionary model test in MEGA X, maximum likelihood was created using the Tamura 3-parameter while in the neighbour-joining method, we used Jukes-Cantor method. We performed evolution analysis using a phylogenetic tree for fourteen *B. africanus* group samples. Seven samples represented four *B. africanus* group samples from Malawi, and seven *Bulinus* were retrieved from the GenBank and included due to their similarities to the project’s sequenced samples. Moreover, *B. succcinoides* and *B. nyassanus* collected from Lake Malawi were included while *B. forskali* from Lake Malawi was included as an out group. The evolutionary analysis was estimated with the bootstrap test (1000 replicates) in MEGA X.

## 3. Results

Generally, analysis of the *B. globosus* shells collected from pond and stream has shown a higher range in size of measurements taken compared to shells of *B. africanus* group collected from the Lake (Table 1). The RFLP assay was utilised as the primary method to classify the species of the individual samples collected based on markers developed in this study. Thus, further investigation was conducted using *cox*1 markers in sequence analysis, resulting in an indication that four species of *B. africanus* group exist within the samples examined, classified based on blast to *B. africanus, B. globosus, B. angolensis* and *Bulinus* sp.

The specific band pattern was consistent in the known samples were identified and verified using DNA sequencing as controls. The rapid essay test RFLP used in this study aimed to distinguish between *B. globosus* and *B. africanus* using 250 samples, including twenty *B. truncatus* samples to test the sensitivity and specificity of the primers used in PCR and markers used in RFLP. As a result, the selected primers used to amplify the Folmer region worked successfully with *B. africanus,* while in *B. truncatus* samples, only a few were amplified. Despite the high level of variation among samples, there was no overlap in the results of *B. globosus, B. africanus* and *B. truncatus* samples. Interestingly, the RFLP assay unexpectedly revealed another group with no cut site: we found *B. angolensis* through further sequence analysis; there is no prior report of finding this species in Malawi. Plus, another group of *B. africanus* group was encountered via the molecular identification.

### 3.1. PCR-Restriction Fragment Length Polymorphism Assay

The PCR-RFLP gel result is illustrated in Figure 3 shows that the *B. globosus* profile was two bands digested by *Sac1* enzyme at about 400 and 200 bp, while *B. africanus* shows two bands at approximately 320 bp. The test confirmed the validity of the developed markers besides the specificity of the selected primers being utilised for *B. africanus* group population. For example, 20 samples from *B. truncatus/tropicus* group from different locations were used as non-*B. africanus* group. Out of the 20 samples, 6 samples illustrate four bands while no amplification was found in 14 samples. Interestingly, this test reveals another species that shows no site cut, which has led to further molecular investigation for all specimens using sequence analysis and evolutionary analysis using phylogenetic tree inferences.

We used 1% agarose gel stained with SYBR™ Safe DNA showing the ‘Folmer’ region of cox1 restricted with double digestion enzymes *Sac1* and *HaeIII*. Known controls of uncut DNA, *B. africanus* (A) and *B. globosus* (G), were used with hyperladder 100 bp as control. The uncut sample (UNCUT) and non-B. *africanus* group (N) are also labelled.

The mitochondrial *cox*1 gene sequence generated for all *B. africanus* group samples in this study was found highly variant, in agreement with a similar study [7,13]. Alignment of 50 sequence samples in this study has revealed four taxa of *B. africanus* group population in Malawi.

### 3.2. Distribution of B. africanus Group Snails on Lake Malawi

Although the surveys were carried out in two different seasons, *B. globosus* which was generally bigger in the shell was not found in the Lake, while all the other three main groups were collected from the Lake. *Bulinus globosus* snails were observed only at a stream, river and pond in Mangochi, Blantyre and Chikhwawa Districts, respectively. In contrast, *B. africanus* snail was found in all visited sites in the Lake whereas *B. angolensis* were collected exclusively from three sites in the Lake, including one site visited in the eastern region (Figure 4).

### 3.3. The Prevalence of Schistosoma DNA within B. africanus Group

A total of 30 of the 106 *Bulinus* snails (28.3%) were positive to the *Schistosoma*-specific screen using real-time PCR methods. DNA *Schistosoma* were detected in *B. africanus* and *B. angolensis* samples in eastern and southern shores of the Mangochi District. The positives samples were collected from the Lake and connected waterbodies.

### 3.4. Phylogenetics Analysis

The maximum likelihood and neighbour-joining trees of *cox*1 gene generated strongly supported phylogenies that showed the presence of four species within the *Bulinus africanus* group in Lake Malawi in addition to two species from the *B. truncatus* complex and *B. forskali* group (Figure 5). The phylogenetic trees illustrate similar results of genetic differentiation of *B. africanus* groups samples. In the *cox*1 tree, the *B. africanus* group specimens are split into four groups, namely *B. africanus*, *B. globosus, B. angolensis* and *Bulinus* sp.

### 3.5. The Differences between Molecular and Morphological Identification

Figure 6 generally shows a problematic identification based on the shell morphology among all four groups. Of note, the *B. globosus* (Figure 6A) from Mangochi, Blantyre and Chikhwawa District populations are very similar by visual inspection of the shell despite far geographical distances and different environmental conditions. Moreover, the *B. globosus* shell was relatively higher than the other groups, which are very similar and overlap in obtained shell measurements.

Although the species of the *B. africanus* group seemed to be similar, a closer investigation demonstrated a subtle morphological distinction in *B. globosus* only: for example, a curved columellar edge, a basal margin join, and a more pronounced and distinctive notch, or ‘tooth’,on the inside *B.*
*globosus* shell; in other species, most of the shells show no extra curvature in the basal margin, and the notch is absent (Figure 6 and Figure 7).

## 4. Discussion

Due to the scale and spread of snails and their contribution to schistosomiasis transmission, updating information and monitoring is important to understand the disease and transmission dynamic. For example, detection of the intermediate host of *S. mansoni* in 2017 has led to increased disease surveillance and revealed the emergence of intestinal schistosomiasis in Mangochi District in 2018, followed by an outbreak of intestinal schistosomiasis in primary schools in Mangochi District in 2019 [16,17]. Additionally, detecting schistosomes hybrid in the same region is a significant epidemiology change and reveals potential new facts about schistosomiasis changes occurring in Lake Malawi (2).

During the past decades, the identification of natural populations and species belonging to the *B. africanus* group has been well known as problematic and sometimes indecisive [8,12]. Although the attempts were made to use some morphology characters such as microsculpture of the shell, it was sometimes misleading rather than helpful [18,19]. However, the combination of morphological and molecular data analysis conducted in this study suggested that at least two separate medical important species from *B. africanus* group species are present in Mangochi District and need to be assessed through a taxonomic appraisal. To our knowledge, this is the first investigation that thoroughly focuses on *B. africanus* group population from Malawi at a molecular level.

With regard to morphology, the *B. globosus* shells in Malawi can be distinguished from *B. africanus* shells by the distinctive small notch or tooth in the truncated columella margin of the aperture inside (Figure 7). A striking finding is that all molecular identification used in this study reveal that *B. africanus*, the intermediate host for *S. haematobium* and *S. mattheei,* is present in Lake Malawi; this species has not been reported before in Malawi. We validated the identification of *B. africanus* by sequence analysis, PCR-RFLP and phylogenetic tree. Finding a new intermediate host in Lake Malawi is considered a new significant alteration in the epidemiology of schistosomiasis transmission in Lake Malawi. The detection of DNA schistosomes in real-time PCR screening test in *B. africanus* is expected as it is known as an intermediate host for schistosomiasis. In contrast, detection of DNA schistosomes in *B. angolensis* was unexpected because *B. angolensis* has not been reported as an intermediate host for schistosomiasis. Therefore, a future study is suggested to investigate the roles of the *B. africanus* group in schistosomiasis in Lake Malawi.

Two possible explanations of the colonisation of *B. africanus* observed in 2017 and 2018 in Lake Malawi might be suggested. The colonisation might have recently occurred, similar to the colonisation of *Biomphalaria,* which was observed in the same surveys [16]. The second scenario is presumed to be because of mistaken identity for *B. globosus* in Lake Malawi in previous studies in light of no report of using molecular identification in this population. Additionally, *B. africanus* was not considered to go beyond the immediate southern African countries. Hence, the molecular applications used in this study, such as blast, sequence analysis and PCR-RFLP rapid essay, sustains the need to develop molecular applications to identify the *B. africanus* group, particularly in distinguishing between *B. globosus* and *B. africanus*.

Molecular identification using *cox*1 gene is a popular method for freshwater snails, as is species identification through sequence comparison with available reference data such as GenBank [4]. Furthermore, *cox*1 evolution is rapid enough to help discriminate species [20]. However, the genetic variation in the *cox*1 marker requires a good selection for critical primers, particularly for *Bulinus* [4,7]. The selected primers for the *B. africanus* group and the sequences generated in this study provide information, and data that can be used for comparison in different populations when trying to understand the distribution of *B. africanus* group species and their role in disease transmission.

Sequence analysis of restriction sites suggested that *Sac1* enzyme could be a species-specific marker for *B. globosus* while *Hae*III enzyme is a marker for *B. africanus* in molecular identification. Upon *Sac*1 digestion, only *B. globosus* sequences were cut, whereas *B. africanus*, *B. angolensis* and *Bulinus* sp. remained intact. Besides, upon double digestion with *Sac*1 and *Hae*III, *B. globosus* and *B. africanus* produced a restriction profile where two fragments were generated that were estimated to be approximately 400 bp and 220 bp for *B. globosus* and two bands at a similar size at 300 bp in *B. africanus*. These restriction data show that a stable genetic marker has been found for *B. globosus* from Malawi. It would now be interesting to ascertain whether the *B. africanus* group from different parts of Africa can be similarly differentiated.

Despite the significant increase in *Bulinus* density after the rainy season in 2018, *B. globosus* was not directly present within the Lake during both surveys. Interestingly, the *cox*1 gene sequence generated for *B. globosus* samples collected from a river in Blantyre, a stream in Chikwawa and a stream in Mangochi District were identical through the distance and different environment. These data sustain the belief that *B. globosus* is not present within this part of the Lake, at least in Mangochi District, reflecting environmental factors such as aquatic plants might play a role in its distribution.

The phylogenetic tree in this study used two methods to measure the evolutionary relatedness among the four taxa collected in this study, with two known species from *B. truncatus/tropicus* complex collected from Lake Malawi and reference samples from GenBank from the *B. africanus* group.

The neighbour-joining and maximum likelihood trees inferred an evolutionary relationship between the four taxa groupings within the specimens investigated (Figure 5). Phylogenetic analysis of the *cox*1 has confirmed that *B. africanus* from Lake Malawi can be separated from *B. globosus* collected from different areas in Malawi. Moreover, the tree is further evidence confirming three species from the *B. africanus* group are found in the lake: *B. africanus*, *Bulinus* sp. and *B. angolensis*. The latter could indicate a new species has not been reported before in Lake Malawi, which would probably reflect mistaken identification because of the similarity in shell morphology between *B. angolensis* and *B. globosus*. Our study provides further evidence for recent study that has suggested based on a genetic investigation that *B. angolensis* is more closely related to the *B. africanus* group than originally documented belonging to the *B. truncatus/tropicus* group [4]. However, looking more closely, tree and sequence analyses in our study found that *B. angolensis* from Lake Malawi is closer to *B. africanus* than *B. globosus*. It is worthwhile noting that Brown (1994) made the point that the treatment of *B. angolensis* is unclear partly as the molecular properties were unknown [12].

Interestingly, this study indicated few samples are similar to the *B. africanus* group based on morphology investigation. However, we could not determine the name of the taxa based on sequence analysis. The specimen shares a node in the phylogenetic tree with *B. africanus* and *B. globosus* specimens but is separate from these species. Although no medical report is known yet for *B. angolensis* and *Bulinus* sp., finding these species with evolutionary closeness to the *B. africanus* group species is providing data that deserve further investigation and could open the door for potential medical importance for these two species.

## Figures and Tables

**Figure 1 tropicalmed-07-00195-f001:**
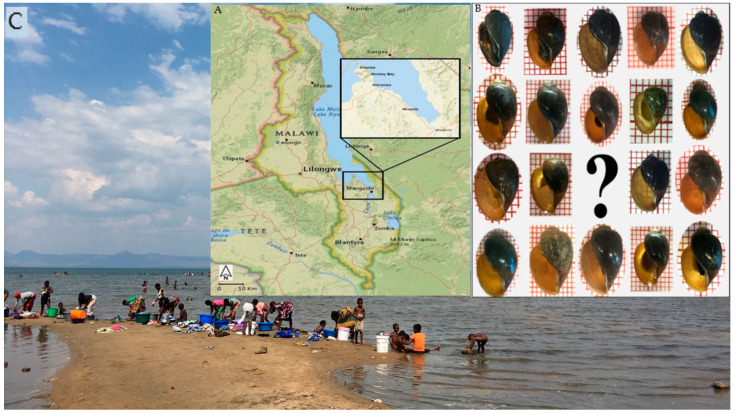
(**A**) Map of Malawi adapted with a highlight of Mangochi District, where the malacological surveys were conducted in 2017 and 2018. (**B**) *B. africanus* shells collected from Lake Malawi were investigated in this study. (**C**) Pictures from locations visited in 2017 and 2018 show the extensive water contact by local communities.

**Figure 2 tropicalmed-07-00195-f002:**
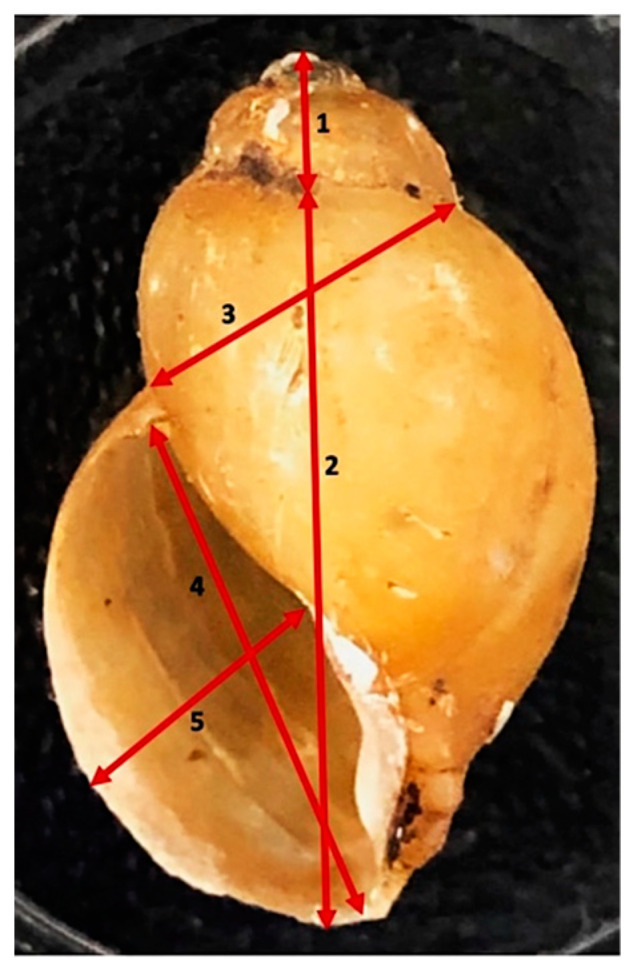
Measurements taken of shells. The picture of a shell shows measurements taken: (1) spire height, (2) height of snail body whorl, (3) snail breadth to body whorl to suture, (4) aperture height, (5) aperture width.

**Figure 3 tropicalmed-07-00195-f003:**
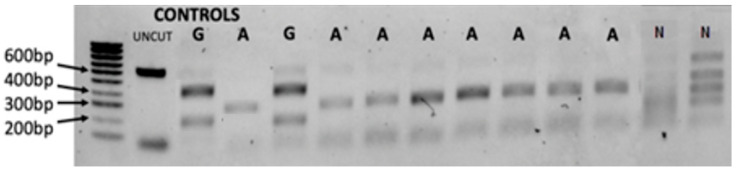
Gel images of the RFLP assay that shows typical PCR-RFLP variation profile.

**Figure 4 tropicalmed-07-00195-f004:**
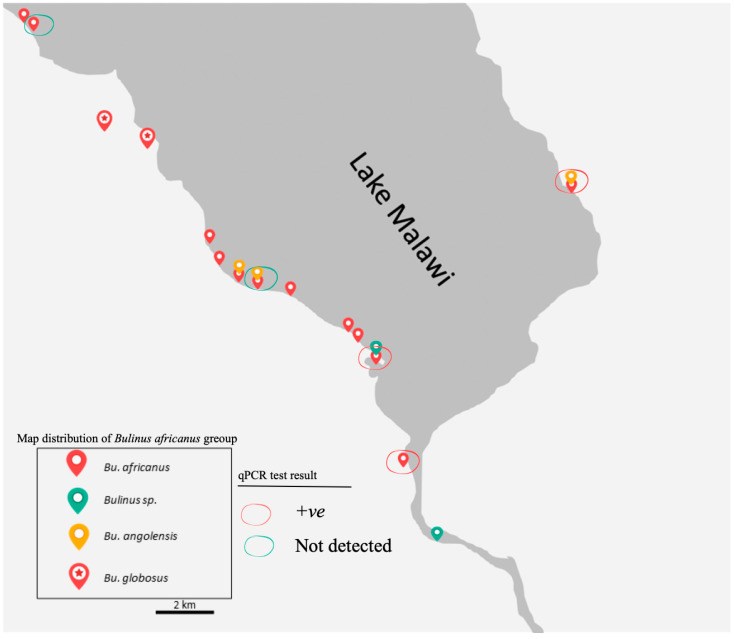
Map of the distribution of the *B. africanus* group on the southern shoreline of Lake Malawi and its connected water bodies visited between 2017 and 2018. The red circle indicates samples were positive (+ve) for real-time PCR, while the green circle indicates samples where DNA of *Schistosoma* were not detected within snail. Previously all snails were thought to be *B. globosus*.

**Figure 5 tropicalmed-07-00195-f005:**
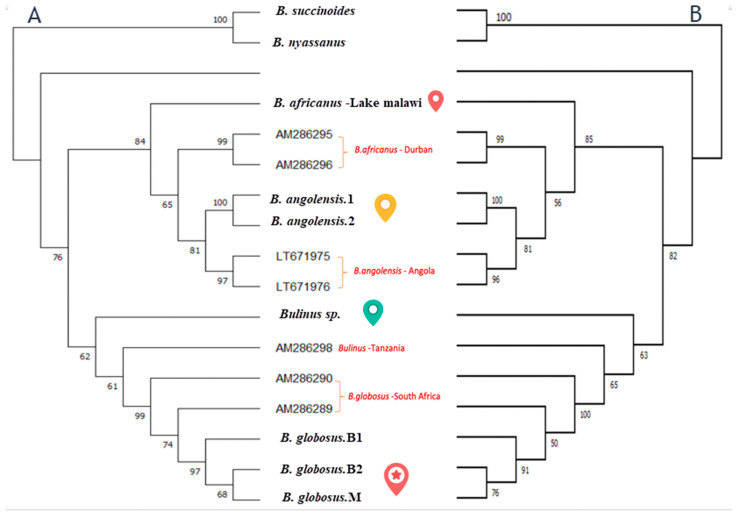
Two phylogenetics trees using maximum likelihood (**A**) and neighbour-joining method (**B**) for *Bulinus* from Lake Malawi. Both the tree of a 467 bp fragment of the (*cox*1) gene for *B. forskali, B. succcinoides, B. nyassanus, B. africanus, B. globosus, B. angolensis* and *Bulinus* sp. (unidentified) collected in this study. Plus, previously published sequences labelled with their accession number in Genbank and collection pint (red). The numbers at the nodes indicate bootstrap proportion based on 1000 replications. The symbols beside the nodes corresponded to specimens that are mentioned above in map distribution.

**Figure 6 tropicalmed-07-00195-f006:**
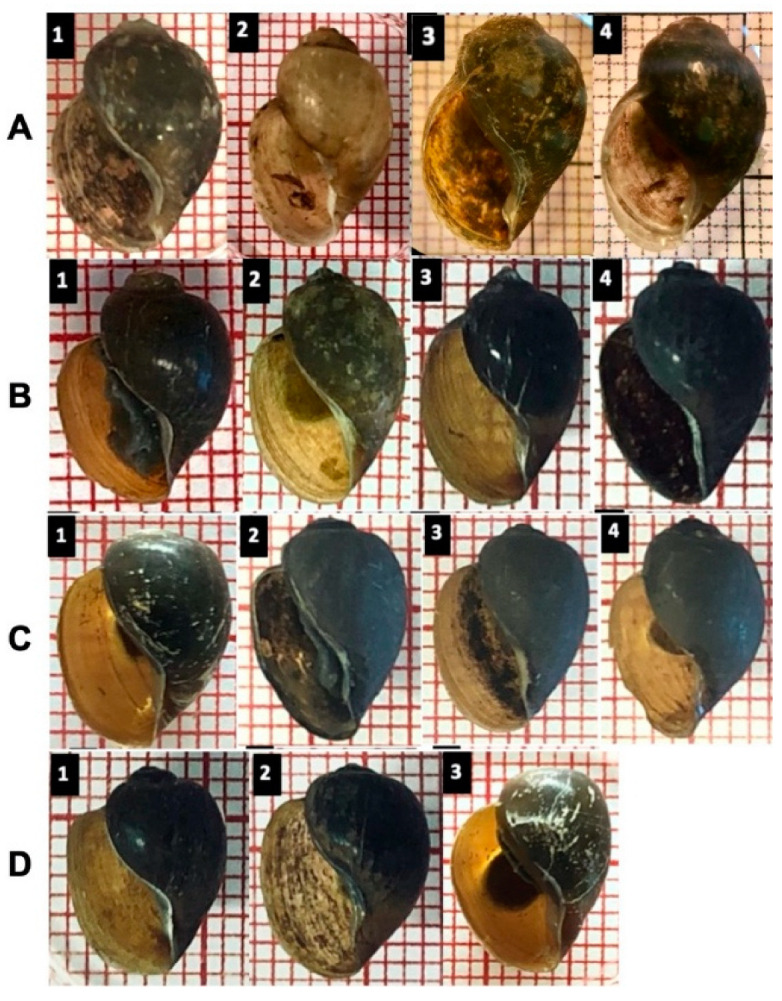
A visual comparison of selected shells indicates some subtle differences. Several shells were identified using both RFLP and sequence analysis. (**A**): *B. globosus* represents three different population habitats; A1–2: from Mangochi District, A3: from Blantyre, A4: Chikhwawa District (**B**): *B. africanus*, (**C**): *B. angolensis*, (**D**): *Bulinus* sp.

**Figure 7 tropicalmed-07-00195-f007:**
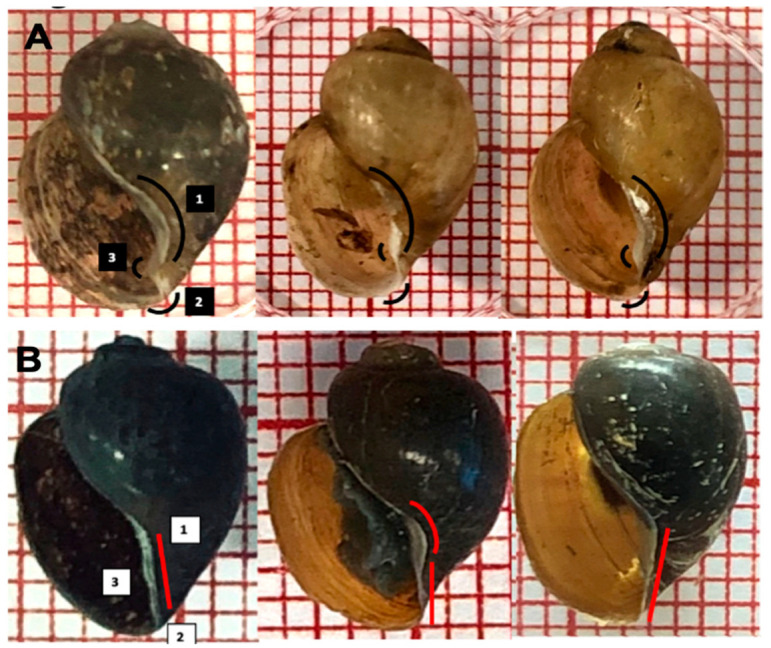
Subtle morphological differences in truncated columella region, a notch or ‘tooth’, in the aperture of *B. globosus.* (**A**): (1) curved columellar edge, (2) basal margin join, (3) distinctive notch on the inside; (**B**): shells and other *B. africanus* group (1 square = 1 mm^2^).

**Table 1 tropicalmed-07-00195-t001:** Summary of the shell measurements for *B. africanus* group.

*B. globosus* (n = 18)	Spire Height (mm)	Lower Body Length (mm)	Whorl-Suture Width (mm)	Aperture Length (mm)	Aperture Width (mm)	Shell Total Height (mm)
Mean (95% CI of the mean)	1.3 (1.1–1.6)	13.2 (12.4–13.9)	5.83 (5.3–6.3)	10.5 (10.0–11.0)	5.0 (4.6–5.3)	14.5 (13.6–15.4)
Std. deviation	0.52	1.5	1.0	1.0	0.7	1.9
*B. africanus* (n = 117)	Spire height (mm)	Lower body length (mm)	Whorl-Suture width (mm)	Aperture length (mm)	Aperture width (mm)	Shell total height (mm)
Mean (95% CI of the mean)	0.7 (0.6–0.8)	9.7 (9.3–10.1)	3.9 (3.7–4.1)	8.1 (7.7–8.4)	3.7 (3.6–3.9)	10.4 (9.9–10.8)
Std. deviation	0.4	2.3	1.1	1.9	0.9	2.5

## Data Availability

The data for this study has been presented within this article and any further information regarding this study can be reasonably requested from the corresponding author.

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
