# Peer review of "First Molecular Identification of *Bulinus africanus* in Lake Malawi Implicated in Transmitting *Schistosoma* Parasites"

_tropicalmed, 2022, doi:10.3390/tropicalmed7080195_

Round 1

Reviewer 1 Report

This report describes the first identification of Bulinus africanus sensu stricto based on molecular identification as an intermediate host for Schistomoma parasites in Lake Malawi where a variety of schistosomiasia is prevalent. The approaches have been well conducted and reports invaluable scientific findings. Thence, this report is suitable for publication in the journal after corrections of many careless mistakes. Please prepare a checked and corrected manuscript of this study for its review process.

Author Response

thanks for your comments, I have corrected the mistakes 

Reviewer 2 Report

Dear Al-Harbi and colleagues,

I have found your work a little bit confusing in several aspects. I would recommend better explain why:

1. The molecular identification of Bulinus genera is necesary and why morphology can not discriminate between B. africanus and B. angolensis.

2. Introduce the current morphological criteria based on biometry or morphology used to identifify Bulinus members

3. You Schistosoma parasites have been identified in B. africanus and B. angolesis that in turn have been collected in different sampling areas. So there is a direct link between Bulinus species and sampling sites, if that is true, why your study is necessary?

4. Your displays should be improved, maps are pixeled as your photos. It would be worth putting a single plate showing the Bulinus species according to their morphological and biometrical criteria to help the reader to understand the need of using molecular tools for taxonomy.

In the attached pdf I have included more comments that I hope will be useful for your research.

Author Response

  1. The molecular identification of Bulinus genera is necesary and why morphology can not discriminate between B. africanus and B. angolensis.

because of the high similarity between the species and also the nature of the shell itself. 

  1. Introduce the current morphological criteria based on biometry or morphology used to identifify Bulinus members

I have added the morphology study to the manuscript

  1. You Schistosoma parasites have been identified in B. africanus and B. angolesis that in turn have been collected in different sampling areas. So there is a direct link between Bulinus species and sampling sites, if that is true, why your study is necessary?

There is only a few info about B. angolesis because it was neglected as it has been believed that it not ply a role in transmission and also it was classified to different group. The finding in this study might open a question about the distribution of this species in Africa particularly with finding hybrid schistosome in the area. On other hand, our study will help in identification of Bulinus species as there is a similarity between Bulinus from trancatus group and b.africanus group. This was also mentioned by (Midsen, et al 2011).

  1. Your displays should be improved, maps are pixeled as your photos. It would be worth putting a single plate showing the Bulinus species according to their morphological and biometrical criteria to help the reader to understand the need of using molecular tools for taxonomy.

I have improved the figure 1 which rise the question about the shell and then in   Figure 5 and 6 I explain the similarity between the investigated samples

In the attached pdf I have included more comments that I hope will be useful for your research.

Thanks for your comments, it was very helpful.

  • africanus group= changed to B. africanus group

I use bu. with bulinus to distingussh from Bi. (biomphalaria)

  • Line 64 - Bicture in figure 1 has been enlarged and also changed .
  • Line 77 – edited
  • I added( +ve )

Reviewer 3 Report

The manuscript submitted is an important contribution for the understanding of the role of Bulinus group in transmitting parasites of the Schistosoma haematobium group in the African region of the Lake Malawi, a crucial spot for endemicity of both host and parasites. In fact the authors demonstrated for the first time that Bu. africanus, the intermediate host for S. haematobium and S. mattheei, is present in Lake Malawi.

Particular attention to identification of parasitized snails should be given in terms of identification of species and of parasitic hybrid forms, as hybridization in biology may underly important features also related to pathogenicity.

In my opinion the manuscript deserves to be published under minor changes, mostly related to evolutionary analyses. In fact, major concerns are about missing information in the phylogenetic inferences and haplotype network analysis, which is included in the results but there is no mention at all in the method section (I suggest the authors to consider to avoid the use of network analysis for separated species, it works better with population of one species because it reveal the diversity among haplotypes that are evolutionary closely related, not distantly related as two separated species can be. (This is the reason why you observed several SNPs along the network branches, sometimes they even appeared separated). Even if evolutionaty analyses are not the main core of the manuscript the info should be improved.

Lines 143-144: Please include more details about the phylogenetic methods you have used. For example: have you tested for the best evolutionary model to be selected in MEGA? have you run the statistical support at nodes with some methods (for example bootstrap)? Have use used retrieved sequences from public repository for comparative purposes? have you used an outgroup? A table with sequences used should be included, even as supplementary material. All these info should be mentioned (as you do after, in the result section, so please move the information in the method section and explain why you used 2 different evolutionary models - Tamura 3-parameter and JC for the two methods if the dataset is the same).

The discussion in lines 340-345 should be mitigated, as nodes with very low statistical support are mentioned (nodes with less than 90 at ML and less than 75 at NJ are not informative and they can be easily be formed by chance and not by relatedness, so please mitigate the conclusion or try to run again the tree including a robust outgroup species to verify the probability of chance, or of some other potential errors outcomes frequently observed in highly variable region as long branch attraction phenomenon).

Minor comments:

line 30, line 54, line 57, line 83, line 85 and along the manuscript: the genus name alone “Bulinus” and “Schistosoma” should be reported in italics

line 31: The group, not capital letter

lines 49-52: RFLP is a useful method if the polymorphisms among taxa to be identified is stable and known (if you intend this with “selected species” I agree, but maybe some additional words to define better selected specie sas “already identified species” or “reference species” should be of help for the reader). This is also about the lines 118-119 “This step was also conducted for selected samples to validate the results from the RFLP rapid assay.”

line 81: “to distinguish between different species in Malawi.” please add of snails or Bulinus species

Figure 1: “where the malacological surveys conducted in 2017, 2018 were conducted” please remove one “conducted”

Line 114: “1 μl of forward and reverse primers”please indicate the concentration of primers not only the volume, in order to let other researchers to repeat the experiment if needed (10ng/ul?)

Line 122: SacI and HaeIII should be in italics (not I and III)

Line 152: “based on blast” should be based in BLAST results indicating identity with…

Line 153 please remove “was consistent”

Line 154 and 293 essay change with assay

Line 166 change SAC1 with SacI

Line 170 “Out of the The”

Line 173 change taxa with specimens and evolution analysis with evolutionary analyses using phylogenetic trees inferences.

Figure 2. Please indicate the meaning of G, A and N.

Line 182 with a similar study – change with similar studies (two references are indicated).

Figure 3: please indicate in the figure which node corresponded to specimens analysed in the present study.

Line 190: Also, to identify the genetic diversity occur in Folmer region. How did you identify the genetic diversity? Have you estimate some diversity parameter as haplotype diversity? Genetic distance? Please indicate it precisely.

Figure 5. Which tree is ML and which NJ? Please indicate the information in the caption.

Line 278: “ITS”??? it is probably a refuse

Be carefull along the manuscript with shcistosomes - schistosomes

Author Response

Thanks for your comments, and suggestions. It was very helpful. 

in terms of creating two trees, It was just a correlation to show that two different trees using two different methods will give the same results. 

I did a lot of changes as required including adding the morphology investigation. I hope the new version of the manuscript is better now. my problem is that your journal asked me to finish the major correction within 10 days which is a very short time for me. 

Round 2

Reviewer 2 Report

You made a good job